# Pre-analytical variables affecting breast cancer biomarker expression: A controlled single-specimen study of fixation duration, cold ischemia time, and fixative preparation in a low-resource setting

Clément Parfait Ndengue[1]*, Paul Jean Adrien Atangana[1,2], Gilbert Roger Ateba[1], Samuel Honoré Mandengue[3], Emile Telesphore Mboudou[1,4], Carole Else Eboumbou Moukoko[2]

1 Douala Gynaeco-Obstetric and Pediatric Hospital (DGOPEH), Douala, Cameroon, 2 Faculty of Medicine and Pharmaceutical Sciences, University of Douala, Douala, Cameroon, 3 Faculty of Sciences, University of Douala, Douala, Cameroon, 4 Faculty of Medicine and Biomedical Sciences, University of Yaounde I, Yaounde, Cameroon

* ndengueparfait@yahoo.fr

## Abstract

### Background

Optimal pre-analytical management of breast tissue specimens, particularly formalin fixation, is essential for accurate immunohistochemical (IHC) biomarker assessment in invasive breast cancer. Although international guidelines suggest using 4% neutral buffered formalin with controlled fixation time, many laboratories in low-resource settings deviate from these standards. This study aimed to determine whether three pre-analytical variables — fixation duration, cold ischemia time, and fixative preparation (4% neutral buffered versus 4% non-buffered formaldehyde) — impact the preservation and evaluation of tissue biomarkers in invasive breast cancer.

### Methods

We conducted an exploratory, proof-of-concept, experimental study using fresh mastectomy tissue from a 34-year-old patient with invasive ductal carcinoma (pT4, hormone receptor-positive, HER2-negative, Ki67 = 40%) who had not received neo-adjuvant chemotherapy. Fifty microsamples (5–15 mm in length, approximately 1 mm in diameter) were obtained using a 14-gauge core needle biopsy device and divided into four cohorts: (1) 19 samples fixed in 4% neutral buffered formaldehyde for 0.5 to 144 hours; (2) 19 samples fixed in 4% non-buffered formaldehyde for 0.5 to 144 hours; (3) 6 samples with delayed fixation (0.5 to 8 hours) then fixed in neutral buffered formaldehyde for 10 hours; (4) 6 samples with delayed fixation (0.5 to 8 hours) then fixed in non-buffered formaldehyde for 10 hours. Hormone receptors (estrogen

**Data availability statement:** All relevant data are within the manuscript and its Supporting Information files. Raw immunohistochemistry scoring data, complete fixation and cold ischemia time parameters, descriptive statistics, and statistical analysis outputs (including the added Spearman correlations and one-way ANOVA across fixation-duration windows performed for this revision) are provided in the S1 Data file.

**Funding:** The author(s) received no specific funding for this work.

**Competing interests:** The authors have declared that no competing interests exist.

**Abbreviations:** ASCO/CAP: American Society of Clinical Oncology/College of American Pathologists, CI: Confidence interval, DGOPEH: Douala Gynaeco-Obstetric and Pediatric Hospital, ER: Estrogen receptor, FFPE: Formalin-fixed paraffin-embedded, HER2: Human epidermal growth factor receptor 2, ICC: Intraclass correlation coefficient, IHC: Immunohistochemistry, IS: Intensity score, LMIC: Low- and middle-income countries, NBF: Neutral buffered formaldehyde, NB-F: Non-buffered formaldehyde, PR: Progesterone receptor, PS: Proportion score, SEM: Standard error of the mean.

receptor-ER, progesterone receptor-PR) and Ki67 expression were evaluated by IHC using the Allred scoring system and current international recommendations.

## Results

Fixative preparation had a statistically significant, yet small, impact on biomarker evaluation. The mean percentage of ER-positive cells was $96.89 \pm 0.74\%$ with neutral buffered formaldehyde compared to $94.32 \pm 1.51\%$ with non-buffered formaldehyde (p = 0.011). Similar trends were seen for PR ($94.89 \pm 0.95\%$ vs. $92.63 \pm 1.67\%$, p = 0.027) and staining intensity. However, Allred scores remained unchanged. Fixation duration was significantly correlated with biomarker expression (Spearman ρ between −0.60 and −0.83, p ≤ 0.007), with stable values from 0.5 to 48 h and a significant decline beyond 72 h (one-way ANOVA across fixation windows: all p < 0.01). Cold ischemia time was strongly correlated with decreased biomarker expression regardless of fixative preparation. Hormone receptor expression and Ki67 remained stable with minimal Allred score changes for up to 2 hours of cold ischemia, but significantly decreased after 2 hours, with scores decreasing in proportion to the duration of ischemia (p < 0.05).

## Conclusions

In this single-specimen controlled experiment, non-buffered formaldehyde preserved tissue biomarkers with small but measurable differences relative to neutral buffered formaldehyde for IHC analysis, although these findings require validation in multi-patient studies. Consistent with current guidelines, a cold ischemia time of up to 1 hour maintained adequate biomarker preservation. These preliminary results may be relevant for pathology laboratories in resource-limited settings where neutral buffered formalin may not be easily accessible, and warrant further investigation across diverse tumor types and baseline expression levels, particularly tumors with ER-low-positive (1–10%) or heterogeneous expression.

## Introduction

Breast cancer remains the most commonly diagnosed cancer and the leading cause of cancer death among women worldwide, with an estimated 2.3 million new cases and 685,000 deaths in 2020 [1]. In sub-Saharan Africa, breast cancer incidence is rising rapidly, with patients often presenting at advanced stages and experiencing poorer outcomes compared to high-income countries [2,3]. Optimal management requires accurate assessment of predictive and prognostic biomarkers, including estrogen receptor (ER), progesterone receptor (PR), human epidermal growth factor receptor 2 (HER2), and Ki67 [4,5], which guide treatment decisions particularly in resource-limited settings where therapeutic options may be constrained.

Immunohistochemistry (IHC) on formalin-fixed paraffin-embedded (FFPE) tissue specimens is the gold standard for biomarker assessment in routine clinical practice

[6,7]. However, the reliability of IHC results critically depends on proper pre-analytical handling, particularly the fixation process [8,9]. Formaldehyde is universally recognized as the optimal fixative for preserving tissue biomarkers [10], and current ASCO/CAP guidelines [4,5] recommend 10% neutral buffered formalin with cold ischemia time (interval between specimen collection and fixation) not exceeding one hour and fixation duration between 6 and 72 hours. Prior foundational work has established that cold ischemia times ≤1 hour are optimal for ER, PR, and HER2 preservation, with detectable deterioration beyond 1–2 hours and progressive loss thereafter [8,11–14].

Despite these well-established recommendations, many pathology laboratories in low- and middle-income countries (LMICs), including those in Cameroon, face significant challenges adhering to these standards. Neutral buffered formalin is not always readily available or affordable, and cold ischemia time is often difficult to control due to logistical constraints, limited infrastructure, and workflow challenges [15,16]. Consequently, laboratories may use non-buffered formaldehyde solutions prepared in-house, and specimens may experience variable delays before fixation.

Previous studies have separately examined the effect of prolonged fixation duration on hormone-receptor and HER2 expression [17–20], the effect of cold ischemia time on the same biomarkers [8,11–14,21], and the effect of different formalin preparations on immunohistochemical assessment of ER, PR, and Ki67 [17,18]. To our knowledge, however, no single controlled experiment has simultaneously varied all three of these pre-analytical variables — fixative preparation (4% neutral buffered versus 4% non-buffered formaldehyde prepared in-house), fixation duration (0.5 h to 144 h), and cold ischemia time (0.5 h to 8 h) — on material from the same invasive breast carcinoma, with a design specifically motivated by the fixative preparations actually available in sub-Saharan African pathology laboratories. The novelty of the present study therefore lies not in any of the individual factors, which are confirmatory, but in their combined, controlled, factorial evaluation on a single tumor in a setting representative of low-resource pathology practice.

In a prior retrospective study at our institution [22], we observed that 85.9% of invasive breast cancer specimens underwent prolonged fixation, with fixation durations reaching up to 83 hours for biopsies and up to 147 hours for mastectomies (mean 88.3±5.4 h for mastectomies), due in large part to weekend and holiday accumulation. This empirical observation motivated the present controlled experimental study, designed to characterize the consequences of the three pre-analytical variables — fixation duration, cold ischemia time, and fixative preparation — across the distribution of fixation durations actually observed in routine practice. We hypothesized that these three variables would significantly influence the preservation and IHC evaluation of tissue biomarkers in invasive breast cancer. We present this work as an exploratory, proof-of-concept experimental study intended to generate hypotheses for subsequent multi-specimen validation.

## Materials and methods

### Ethical considerations

This study was approved by the Institutional Ethics Committee of Douala Gynaeco-Obstetric and Pediatric Hospital (DGOPEH), Cameroon (approval number: N°2024/1617/HGOPED/DG/CEI). Prof. Emile Telesphore MboudoU, Chair of the DGOPEH Institutional Ethics Committee, was not part of the research team at the time of protocol submission and ethics review; he therefore had no conflict of interest with respect to this protocol during its evaluation by the committee. He was subsequently associated with the preparation of the present manuscript in his institutional capacity (Project administration, Resources). The study utilized surplus surgical waste tissue from a mastectomy specimen accessed on 30/03/2023. The specimen was completely anonymized prior to analysis and identified only by laboratory code assigned during routine pathology processing. Authors had no access to information that could identify the participant during or after data collection. Data collection and analysis were performed between March 2023 and March 2026.

### Study design and specimen characteristics

We conducted an exploratory, proof-of-concept controlled experimental study at the Anatomic Pathology Department of DGOPEH, Cameroon, between March and July 2023. The biological material consisted of a fresh mastectomy specimen

with axillary lymph node dissection accessed on 30/03/2023 from a 34-year-old female patient diagnosed with invasive ductal carcinoma of the left breast who had not received neoadjuvant chemotherapy. A single-specimen experimental design was chosen to eliminate inter-tumor biological variability as a confounding factor, consistent with established methodology in pre-analytical fixation research [8,11,17]. Pre-operative core needle biopsy had established moderately differentiated invasive ductal carcinoma, clinical stage pT4, positive for ER and PR, negative for HER2 by IHC, and highly proliferative with a Ki67 index of 40%.

### Experimental design

**Samples collection and fixative preparation.** Using a 14-gauge core needle biopsy device, we obtained 50 microsamples from the tumor mass. Each microcore measured approximately 5–15 mm in length and approximately 1 mm in diameter. Two types of formaldehyde fixatives were prepared:

4% Neutral Buffered Formaldehyde (NBF): Commercial ready to use 4% neutral buffered formaldehyde solution (Q Path™), pH 6.9.

4% Non-Buffered Formaldehyde (NB-F): Prepared in house by diluting commercial formalin (39% w/v formaldehyde; VWR Chemicals) with tap water at a ratio of 9:1 (50 ml formalin + 450 ml tap water), yielding an acidic solution through the reaction: $H\text{-}CHO + H_2O \rightarrow H\text{-}COOH + H_2$

**Experimental groups.** The 50 samples were divided into four experimental cohorts:

Cohort 1 (n = 19): Immediate fixation in 4% NBF with varying fixation durations (0.5, 1, 2, 3, 4, 5, 6, 7, 8, 9, 10, 11, 12, 24, 48, 72, 96, 120, and 144 hours). Samples were immediately placed in labeled 50 ml containers containing 15 ml of 4% NBF.

Cohort 2 (n = 19): Immediate fixation in 4% NB-F with the same fixation durations as Cohort 1.

Cohort 3 (n = 6): Delayed fixation followed by 10 hours fixation in 4% NBF. Samples were kept at room temperature (~25°C) for varying cold ischemia times (0.5, 1, 2, 4, 6, and 8 hours) before fixation.

Cohort 4 (n = 6): Delayed fixation followed by 10 hours fixation in 4% NB-F, with the same cold ischemia times as Cohort 3.

The fixative to tissue ratio was maintained at approximately 15:1 or greater for all samples.

The 0.5–144 h fixation range was deliberately selected to span the distribution of fixation durations actually observed in routine practice at the DGOPEH pathology laboratory. A prior retrospective analysis of 64 invasive breast cancer specimens processed between 2017 and 2020 at the same institution [22] documented that 85.9% of specimens underwent prolonged fixation, with fixation durations of 18–83 hours for biopsies, 19–82 hours for nodulectomies, and 43–147 hours for mastectomies (mean 88.3 ± 5.4 h for mastectomies). Although our 14-gauge microcores are morphologically analogous to biopsies, the 0.5–144 h range tested here was deliberately chosen to span the full distribution of fixation durations observed in routine practice across all specimen types at our institution, thereby providing technical benchmarks applicable to biopsies, nodulectomies, and mastectomies alike. Fixation durations of 72–144 h correspond to specimens held in fixative over weekends or public holidays, a recurring logistical reality in resource-limited settings with limited pathology laboratory staffing. Durations beyond 72 h lie outside the ASCO/CAP-recommended 6–72 h window and are therefore presented here as technical benchmarks characterizing the safety margin of local practice, not as endorsed clinical protocols.

**Tissue processing and immunohistochemistry.** After fixation, all tissue samples underwent standardized manual processing including dehydration through graded alcohols, clearing in xylene, paraffin infiltration, and embedding. Paraffin blocks were sectioned at 5 µm thickness using a Leica manual rotary microtome. H&E stained sections were examined by a board certified pathologist to confirm the presence of invasive carcinoma in all 50 samples.

Immunohistochemical analysis was performed using an automated immunostaining system (BenchMark ULTRA, Ventana Medical Systems, Roche Diagnostics, Tucson, AZ, USA). The protocol included: deparaffinization at 75°C,

heat-induced epitope retrieval using Cell Conditioning Solution 1 (CC1, pH 8.0; Cat. No. 950–124) at 95°C for 64 minutes, endogenous peroxidase blocking, primary antibody incubation for 32 minutes at 37°C, detection with the OptiView DAB IHC Detection Kit (Cat. No. 760–700), and counterstaining with Hematoxylin II (Cat. No. 790–2208). The following ready-to-use rabbit monoclonal primary antibodies (CONFIRM range, Roche Diagnostics) were used at the manufacturer-optimized working concentration of approximately 1 µg/mL: CONFIRM anti-Estrogen Receptor (ER) clone SP1 (Cat. Nos. 790–4324 [50 tests] or 790–4325 [250 tests]); CONFIRM anti-Progesterone Receptor (PR) clone 1E2 (Cat. Nos. 790–2223 [50 tests] or 790–4296 [250 tests]); and CONFIRM anti-Ki-67 clone 30–9 (Cat. No. 790–4286, 50 tests). Appropriate positive and negative tissue controls were included in each immunostaining run, together with a negative reagent control (CONFIRM Negative Control Rabbit Ig, Cat. No. 760–1029).

**Biomarker assessment.** ER and PR expression were evaluated using the Allred scoring system by two independent observers blinded to experimental conditions. The Allred score combines Proportion Score (PS: percentage of positive tumor cells) and Intensity Score (IS: staining intensity 0–3), with total scores of 0–2 considered negative and 3–8 considered positive [4,6]. Ki67 expression was assessed according to International Ki67 in Breast Cancer Working Group recommendations, with tumors classified as low proliferative (<10%), intermediate (10–30%), or high proliferative (>30%) [7]. For all quantitative variables, the final value was calculated as the mean of the two independent observers assessments.

## Statistical analysis

Statistical analyses were conducted using IBM SPSS Statistics version 25.0 and Python (SciPy 1.11). Continuous variables were presented as mean±standard error of the mean (SEM). Spearman's rank correlation coefficient ($\rho$) was used to evaluate the relationship between fixation duration or cold ischemia time and biomarker expression levels. Paired t-tests were utilized to compare mean biomarker expression levels between the two fixative types, with 95% confidence intervals (CI) calculated for mean differences. To formally test the effect of fixation duration, we also performed one-way ANOVA comparing four a priori fixation-duration windows (0.5–12 h, 24–48 h, 72–96 h, 120–144 h) separately for each fixative and each biomarker, with Welch pairwise t-tests between the reference window and the other windows. Two-way ANOVA was performed to assess interaction effects. A two-tailed p-value<0.05 was considered statistically significant. Interobserver variability was evaluated by calculating the intraclass correlation coefficient (ICC). We note that, given the single-specimen design, reported p-values describe within-specimen technical variation under controlled pre-analytical conditions and do not support population-level inference.

## Results

### Sample characteristics and quality control

All 50 samples contained tumor tissue as confirmed by H&E staining. The ICC between the two independent observers was 0.94 for ER, 0.92 for PR, and 0.96 for Ki67 evaluation, indicating excellent inter observer agreement. Under optimal conditions (immediate fixation in 4% NBF for 0.5–12 hours), the tumor exhibited strong biomarker expression: ER 96–100% positive cells (Allred score 7–8), PR 95–100% positive cells (Allred score 7–8), and Ki67 38–40% (high proliferative).

### Effect of fixative preparation on biomarker expression

When comparing samples fixed in 4% NBF versus 4% NB-F across all fixation durations (0.5–144 hours), statistically significant differences were observed in both the percentage of positive cells and staining intensity for ER and PR (Table 1).

Despite statistical significance, Allred scores remained unchanged between the two fixative types for fixation durations up to 48 hours. Both fixatives showed progressive decline in expression after 48 hours (Fig 1).

**Table 1. Comparison of biomarker expression between fixative types.**

| Variable | NBF (mean±SEM) | NB-F (mean±SEM) | Mean difference | 95% CI | p value |
|---|---|---|---|---|---|
| ER, % positive cells | 96.89±0.74 | 94.32±1.51 | 2.58 | 0.67–4.49 | 0.011 |
| ER, staining intensity | 2.82±0.07 | 2.63±0.11 | 0.18 | 0.02–0.35 | 0.031 |
| PR, % positive cells | 94.89±0.95 | 92.63±1.67 | 2.26 | 0.29–4.24 | 0.027 |
| PR, staining intensity | 2.74±0.10 | 2.53±0.10 | 0.21 | 0.06–0.36 | 0.007 |
| Ki67, % positive cells | 34.33±3.02 | 30.83±4.77 | 3.50 | −1.08 to 8.08 | 0.107 |

NBF, Neutral buffered formaldehyde; NB-F, Non-buffered formaldehyde; SEM, Standard error of the mean; CI, Confidence interval.

### Effect of fixation duration on biomarker expression

For samples with immediate fixation (no cold ischemia), biomarker expression remained optimal and stable from 0.5 to 48 hours for both fixatives, with Allred scores remaining at 7–8 for hormone receptors and Ki67 at 38–40%. After 48 hours, gradual decline was observed, with more pronounced decrease after 72 hours (Fig 2). Spearman rank correlations between fixation duration (0.5–144 h) and percentage of positive cells were statistically significant for all three biomarkers in both fixatives: ER — NBF $\rho = -0.736$ (p=0.0003), NB-F $\rho = -0.658$ (p=0.0022); PR — NBF $\rho = -0.738$ (p=0.0003), NB-F $\rho = -0.830$ (p<0.0001); Ki67 — NBF $\rho = -0.596$ (p=0.007), NB-F $\rho = -0.646$ (p=0.003). One-way ANOVA across fixation-duration windows (0.5–12 h, 24–48 h, 72–96 h, 120–144 h) was significant for all biomarkers in both fixatives: ER — NBF F=22.96 (p<0.0001), NB-F F=25.63 (p<0.0001); PR — NBF F=28.99 (p<0.0001), NB-F F=45.48 (p<0.0001); Ki67 — NBF F=5.93 (p=0.007), NB-F F=6.13 (p=0.006). Pairwise Welch t-tests indicated that the 24–48 h window was not significantly different from the 0.5–12 h reference window (all p>0.15), whereas the 120–144 h window differed significantly for ER and PR percentages in both fixatives. At extended fixation durations (Table 2), progressive deterioration occurred with both fixatives.

### Effect of cold ischemia time on biomarker expression

Cold ischemia time demonstrated strong negative correlation with biomarker expression, regardless of fixative type (Fig 3). For ER, expression remained near baseline (95–97% positive cells, intensity 2.5–3.0) at 0.5–1 hour, showed noticeable decline at 2 hours (79–83% positive, intensity 2.0–2.5), significant decrease at 4 hours (61–63% positive, intensity 1.5–2.0), marked reduction at 6 hours (41–52% positive, intensity 1.0–1.5), and severe decline at 8 hours (33–35% positive, intensity 1.0). Spearman correlations were $\rho = -0.95$ (p<0.001) for NBF and $\rho = -0.97$ (p<0.001) for NB-F.

PR expression demonstrated similar sensitivity with even steeper decline: baseline at 0.5–1 hour (95–97% positive, intensity 2.5–3.0), decline to 80–81% at 2 hours, 59–66% at 4 hours, 47% at 6 hours, and 24–31% at 8 hours (some becoming negative). Spearman correlations were $\rho = -0.96$ (p<0.001) for NBF and $\rho = -0.98$ (p<0.001) for NB-F.

Ki67 expression also declined significantly: 38–40% at 0.5–1 hour, 38–39% at 2 hours, 33–35% at 4 hours, 27–31% at 6 hours, and 9–21% at 8 hours. Spearman correlations were $\rho = -0.94$ (p<0.001) for NBF and $\rho = -0.96$ (p<0.001) for NB-F.

Analysis revealed critical time thresholds: ≤1 hour showed no significant Allred score changes; 2 hours showed borderline changes but most samples remained positive with slightly reduced scores; >2 hours showed progressive and clinically significant decline with potential for false-negative results.

Two-way ANOVA revealed cold ischemia time as the dominant factor (F=87.3, p<0.001), with no significant effect of fixative type (F=2.1, p=0.156) or interaction (F=0.8, p=0.523).

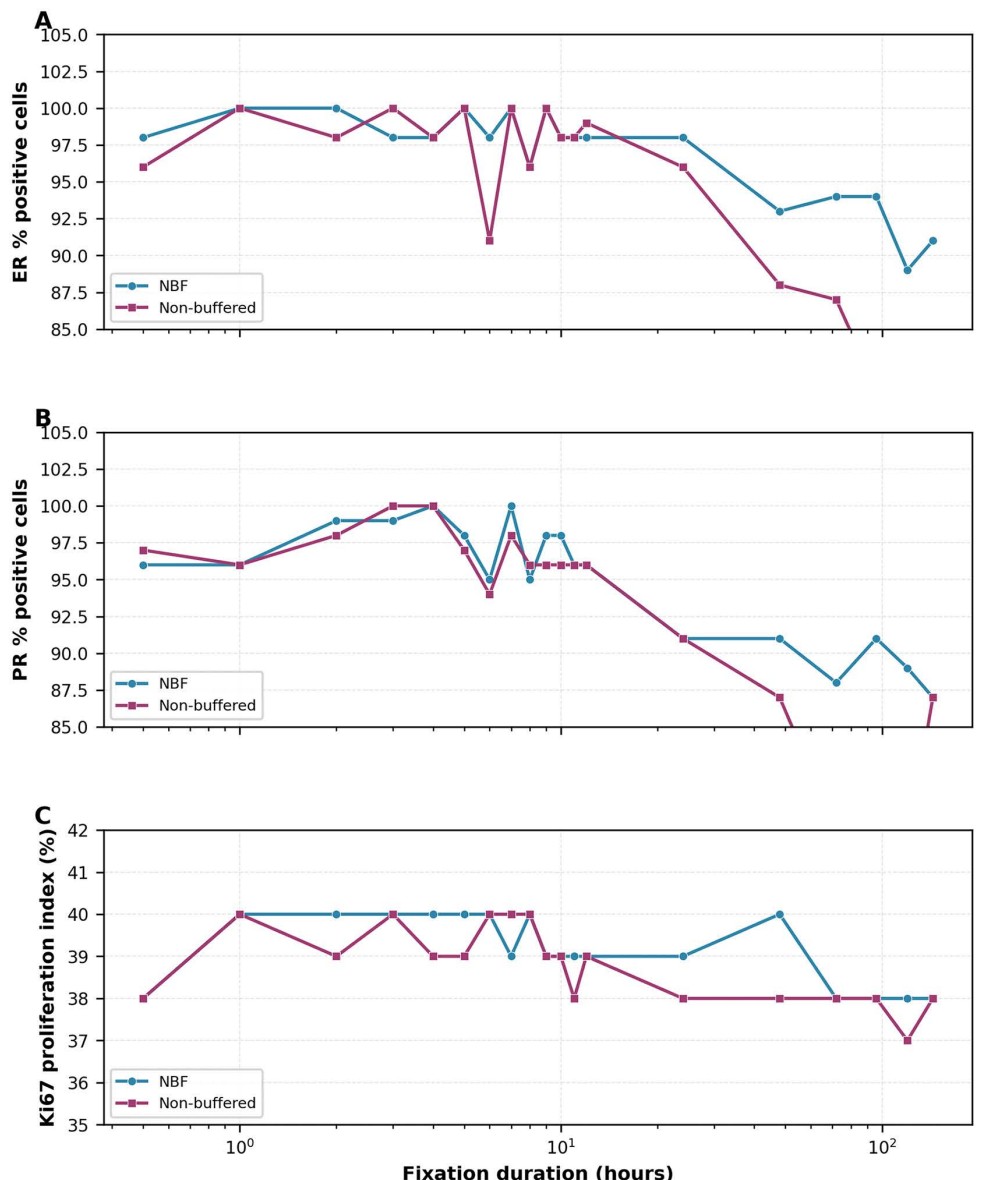

**Fig 1. Effect of fixation duration on biomarker expression in invasive breast cancer. (A)** Estrogen receptor (ER) percentage of positive cells. **(B)** Progesterone receptor (PR) percentage of positive cells. **(C)** Ki67 proliferation index. In each panel, samples fixed in 4% neutral buffered formaldehyde (NBF, blue) are overlaid with samples fixed in 4% non-buffered formaldehyde (NB-F, magenta). Data points represent mean values from duplicate observations by two independent evaluators. Note logarithmic x-axis scale. Biomarker expression remained stable up to 48 hours, followed by progressive decline with extended fixation (72–144 hours). Both fixative types showed similar preservation patterns, with slightly higher values for NBF (p<0.05).

## Discussion

This exploratory, proof-of-concept experimental study investigated the combined effects of three pre-analytical variables — fixation duration, cold ischemia time, and fixative preparation — on tissue biomarker preservation in a single invasive breast cancer specimen with strong baseline expression. The single-specimen design controls inter-tumor biological variability and allows direct attribution of observed within-specimen changes to experimental conditions, but it cannot license

**A**

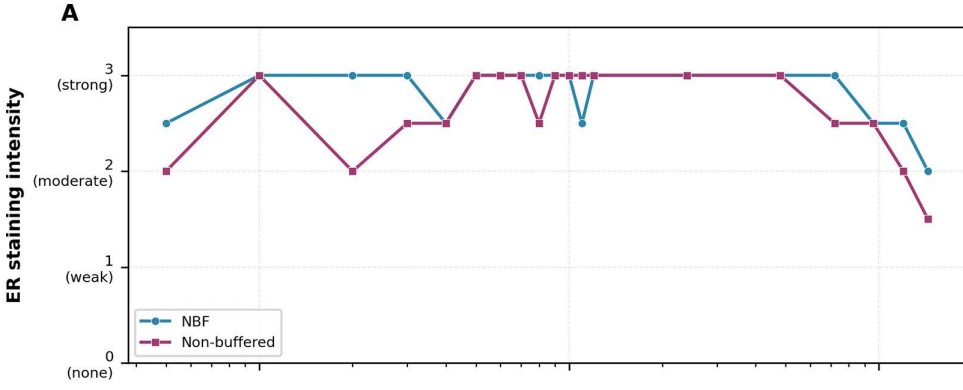

**B**

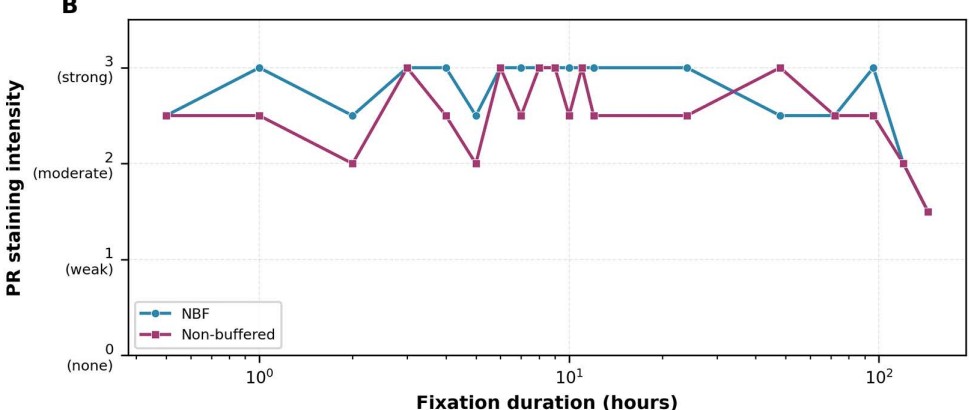

**Fig 2. Effect of fixation duration on staining intensity. (A)** Estrogen receptor staining intensity. **(B)** Progesterone receptor staining intensity. In each panel, NBF (blue) and NB-F (magenta) are overlaid. Intensity scores: 0 = none, 1 = weak, 2 = moderate, 3 = strong. Both fixatives maintained strong to moderate intensity (scores 2.5–3.0) for hormone receptors up to 48 hours, with gradual decline to moderate intensity (scores 1.5–2.0) at extended fixation times (120–144 hours).

**Table 2. Biomarker expression at extended fixation times (ER %/ PR %/ Ki67%, NBF vs NB-F).**

| Fixation time | ER % NBF | ER % NB-F | PR % NBF | PR % NB-F | Ki67% NBF | Ki67% NB-F |
|---|---|---|---|---|---|---|
| 48 hours | 93 | 88 | 91 | 87 | 40 | 38 |
| 72 hours | 94 | 87 | 88 | 78 | 38 | 38 |
| 96 hours | 94 | 81 | 91 | 80 | 38 | 38 |
| 120 hours | 89 | 83 | 89 | 77 | 38 | 37 |
| 144 hours | 91 | 83 | 87 | 87 | 38 | 38 |

population-level inference. The findings presented here should therefore be interpreted as hypothesis-generating and require validation across multiple tumors, particularly tumors with lower or heterogeneous baseline expression, before any change in clinical practice is considered. Within these constraints, our results suggest that fixative preparation had a small statistical effect without categorical reclassification in this index tumor, while cold ischemia time emerged as the dominant pre-analytical variable affecting biomarker preservation.

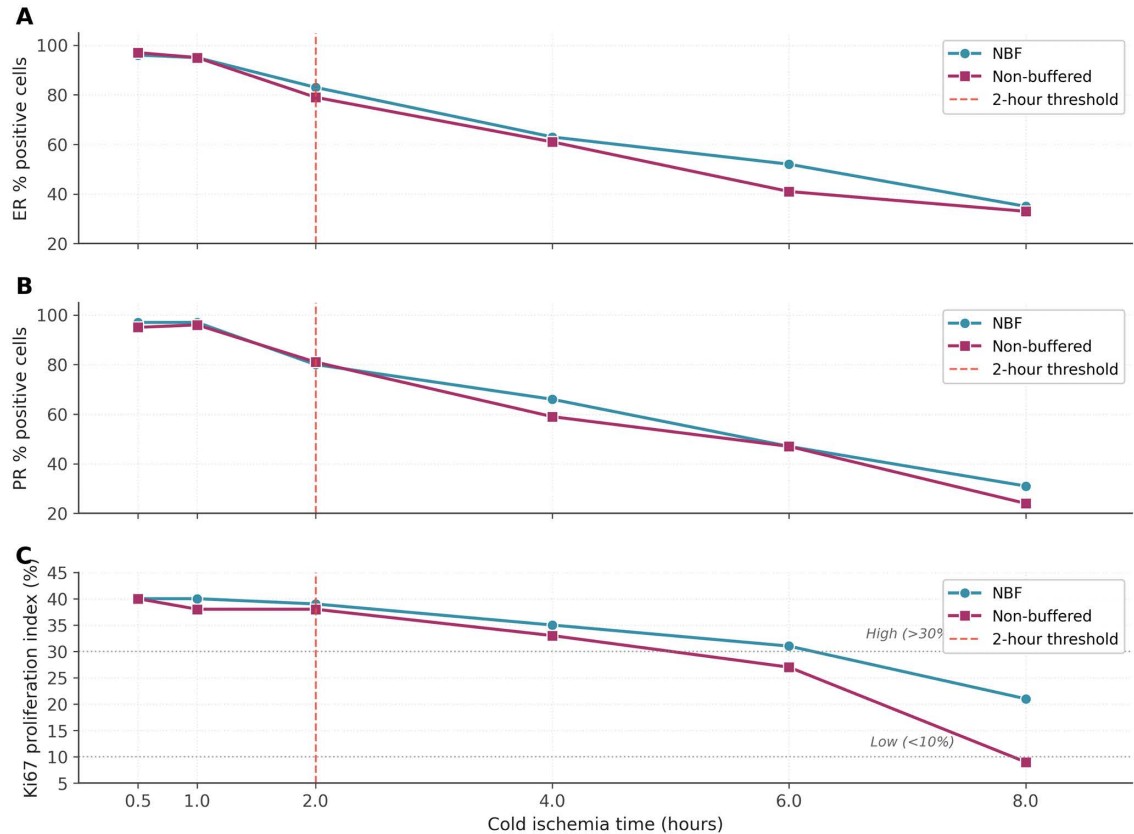

**Fig 3. Effect of cold ischemia time on biomarker expression. (A)** Estrogen receptor percentage of positive cells. **(B)** Progesterone receptor percentage of positive cells. **(C)** Ki67 proliferation index. In each panel, samples fixed in NBF (blue) and NB-F (magenta) after the indicated cold ischemia time are overlaid; the dashed vertical line marks the 2-hour threshold beyond which Allred scores began decreasing significantly. Strong negative correlations were observed for all biomarkers (Spearman ρ ≤ −0.94, p<0.001). Biomarker expression remained adequate (>90% for hormone receptors, 38–40% for Ki67) up to 1 hour. Noticeable decline occurred at 2 hours, with dramatic deterioration beyond 4 hours, regardless of fixative type. All samples underwent standardized 10-hour fixation after the specified cold ischemia time. NBF, Neutral buffered formaldehyde; NB-F, Non-buffered formaldehyde.

## Principal findings and fixative preparation

We found three main results. First, formaldehyde preparation type (neutral buffered versus non-buffered) showed statistically significant but small differences in this index tumor. Although 4% NBF demonstrated slightly superior preservation of ER (mean difference 2.58%, p=0.011) and PR (mean difference 2.26%, p=0.027) compared to 4% NB-F, these differences did not translate into Allred score changes or clinical classification changes in our strongly ER- and PR-positive index tumor. We emphasize, however, that these differences are of a magnitude that could cross the diagnostic threshold in tumors with weak baseline expression (e.g., ER-low-positive tumors with 1–10% positive cells, as defined in the ASCO/CAP 2020 update [4]), and our data therefore do not support a general claim of equivalence between the two fixative preparations. Second, both fixative types adequately preserved biomarkers for 0.5 to 48 hours, after which progressive deterioration occurred. Third, cold ischemia time emerged as the dominant pre-analytical factor, with a critical threshold of approximately 2 hours beyond which clinically relevant biomarker degradation occurred regardless of fixative type. Although each of these effects individually replicates prior work, the contribution of the present study lies in their joint evaluation under controlled conditions on material from a single tumor.

The minimal difference between neutral buffered and non-buffered formaldehyde is somewhat counterintuitive given pH-controlled fixation's theoretical advantages. Formaldehyde fixation proceeds through complex reactions involving formaldehyde addition to amino groups, followed by dehydration and methylene bridge formation [10]. While neutral buffered formaldehyde at pH 6.8–7.2 is optimal for initial addition, non-buffered formaldehyde (acidic through formic acid formation: $H-CHO + H_2O \rightarrow H-COOH + H_2$) may facilitate the dehydration step more efficiently due to lower pH (~4.0–5.5) [10,23].

Our findings align with Apple et al., who found different fixative types yielded comparable hormone receptor results to standard neutral buffered formalin [17], and Oyama et al., who reported fixative type did not significantly influence ER assessment with automated IHC protocols [18]. Our study extends these findings by specifically comparing the most commonly available formaldehyde preparations in resource-limited settings and demonstrating that statistical differences do not translate into clinically meaningful changes in a strongly ER/PR-positive tumor.

This finding is particularly relevant for pathology laboratories in LMICs where neutral buffered formalin may be expensive, unavailable, or subject to supply chain disruptions. Our results suggest laboratories using properly diluted non-buffered formaldehyde (prepared from commercial 37–40% formalin) can achieve adequate biomarker preservation for routine IHC analysis, provided other pre-analytical variables, particularly cold ischemia time, are carefully controlled, and that this interpretation is confirmed in multi-specimen studies across diverse tumor types.

## Effect of fixation duration

Biomarker expression in our specimen remained stable throughout the 0.5–48 h fixation window and declined progressively at extended durations, with statistically significant one-way ANOVA effects across fixation windows (all $F \geq 5.93$, $p \leq 0.007$) and strong negative Spearman correlations between fixation duration and percentage of positive cells ($\rho$ between −0.60 and −0.83, $p \leq 0.007$). The stability observed up to 48 h is consistent with the ASCO/CAP-recommended 6–72 h window [4,5] and with prior systematic reviews showing that hormone receptor and HER2 expression are well preserved within this interval [19]. The progressive decline observed beyond 72 h, most marked at 120–144 h, is in line with earlier experimental reports of gradual antigen loss with prolonged formalin exposure [11,12] and likely reflects over-cross-linking of protein epitopes and subtle tissue-matrix changes that compromise immunoreactivity. Notably, PR percentages declined earlier and more steeply than ER percentages, whereas Ki67 was relatively preserved across the full 0.5–144 h range — a pattern consistent with the differential protein stability of these three antigens under prolonged fixation. The clinical relevance of these observations is that fixation durations of 72–144 h, although outside the ASCO/CAP-recommended window, are empirically observed in sub-Saharan African pathology laboratories — particularly for mastectomy specimens, which routinely reach 43–147 h of fixation at our institution, but also for biopsies, which may reach up to 83 h [22], when specimens are held in fixative over weekends or public holidays. Our data suggest that in strongly-positive tumors such occasional prolongation may not change categorical classification, but also that this reassurance may not extend to tumors with weaker baseline expression, where the observed absolute decline could cross the diagnostic threshold.

## Critical impact of cold ischemia time

The most clinically significant finding is cold ischemia time's profound negative impact on biomarker preservation. We observed strong negative correlations (Spearman $\rho \leq -0.94$, $p < 0.001$) between cold ischemia time and all tested biomarkers, with dramatic effects beyond 2 hours. At 8 hours, PR expression declined to borderline negative levels (24–31%), and Ki67 dropped from high proliferative (38–40%) to low proliferative (9–21%) ranges. These changes have direct clinical implications, potentially leading to inappropriate treatment decisions such as withholding endocrine therapy or underestimating tumor aggressiveness, a concern recently echoed in the context of post-neoadjuvant breast cancer specimens [21].

Our findings partially contradict Apple et al. [17], but this discrepancy may be explained by differences in initial biomarker expression levels. Several studies demonstrated that cold ischemia time's impact is inversely related to baseline

expression intensity [13,14]. Tumors with strong initial positivity (>80% cells with strong intensity), like our specimen, may tolerate brief fixation delays with minimal classification changes. However, tumors with moderate or weak baseline expression (10–50% positive cells or weak-moderate intensity) are more vulnerable to cold ischemia-induced degradation and may shift from positive to negative status with relatively short delays [11,12].

The mechanistic basis relates to continued tissue protease and degradative enzyme activity following tissue excision. While tissue no longer receives blood supply, cellular metabolism and enzymatic activity continue for hours, particularly for abundant enzymes such as proteases, phosphatases, and nucleases [9]. Formaldehyde fixation rapidly inactivates these enzymes by protein cross-linking, "freezing" the tissue's molecular state. Delayed fixation allows progressive enzymatic antigen degradation, with particularly rapid effects on labile proteins [11,24].

Our observation that PR showed greater cold ischemia sensitivity than ER is consistent with previous reports [12] and has important clinical implications. PR expression is considered a functional indicator of ER pathway integrity, and ER+/PR- tumors may represent a distinct biological subtype with different clinical behavior [25]. Preferential PR loss with delayed fixation could lead to misclassification of true ER+/PR+ tumors as ER+/PR-, potentially affecting treatment decisions and prognostic assessments.

## Practical implications and study limitations

If confirmed in multi-specimen studies, our findings would have potentially important practical implications for pathology practice in resource-limited settings. First, they would suggest that laboratories lacking access to commercial neutral buffered formalin may be able to use properly prepared non-buffered formaldehyde without substantially compromising hormone receptor and Ki67 assessment quality in strongly-positive tumors, pending validation in tumors with borderline expression. Such a finding could help reduce costs and improve diagnostic service sustainability in low-resource environments, but should not be adopted as practice without prospective multi-specimen validation.

Second, our data are consistent with current ASCO/CAP guidelines [4,5] and underscore the critical importance of minimizing cold ischemia time. Current best practice guidelines recommend fixation within 1 hour [4,5], and our results are consistent with this recommendation. Practical strategies to minimize cold ischemia time in resource-limited settings — all of which are already supported by existing ASCO/CAP guidelines rather than novel recommendations arising from our data — include: providing fixative containers directly in operating rooms; training surgical staff to place specimens immediately into fixative; establishing efficient specimen transport systems; implementing quality assurance programs tracking cold ischemia times; and including cold ischemia time information on pathology reports to aid result interpretation. These interventions require minimal financial investment but significant attention to workflow optimization and staff education.

Study limitations should be carefully considered when interpreting these results. Most importantly, we studied a single tumor specimen with strong baseline hormone receptor expression (ER 100%, PR 95–100%) and high Ki67 (40%). This single-specimen design, while standard in controlled pre-analytical studies and advantageous for eliminating inter-tumor variability as a confounding factor [8,17], inherently limits the generalizability of our findings and raises concerns of pseudoreplication: the 50 microcores, although fixed and processed independently, derive from a single biological donor and therefore do not provide independent estimates of population-level effects. The p-values reported should be read as measures of within-specimen technical variation under controlled conditions, not as licenses for population-level inference. The index tumor exhibited strong, diffuse ER and PR positivity and high Ki67, placing it far from the diagnostic decision thresholds (1% for hormone-receptor positivity, with the 1–10% "ER-low-positive" range now defined in the ASCO/CAP 2020 update [4]; 10% and 30% for Ki67 proliferation categories). In this setting, even biologically meaningful declines in staining are unlikely to shift categorical classification. Tumors with lower or heterogeneous baseline expression — particularly those with weak (1–10%) ER or PR positivity, focal or mosaic staining patterns, or Ki67 near the 10–30% range — are intrinsically closer to these thresholds and may therefore be reclassified by the same absolute magnitude of pre-analytical loss observed here. Intra-tumoral heterogeneity of ER, PR, and

Ki67 is well documented in invasive breast carcinoma and was not systematically addressed by our sampling strategy; the 50 microcores were drawn from different regions of the tumor mass but may nevertheless have been enriched for regions with uniformly strong expression. Tumors with lower or moderate baseline expression, which are closer to clinically relevant diagnostic thresholds, may show greater sensitivity to pre-analytical variables and could yield different results; therefore, our observations should not be extrapolated to all breast cancer subtypes without further validation. Multi-patient studies incorporating tumors with diverse molecular subtypes, varying baseline expression levels (particularly tumors with borderline positivity), and different histological grades are needed to confirm and extend these findings before any practice recommendations can be made. The study was performed at room temperature (~25°C) typical for tropical climates; cold ischemia effects may differ at different ambient temperatures. We assessed only three biomarkers (ER, PR, and Ki67); other markers, including HER2, may show different sensitivities. Our assessment was limited to immunohistochemistry; effects on other analytical methods such as fluorescence in situ hybridization or molecular assays may differ. While we demonstrate technical impact on biomarker staining, we did not assess ultimate clinical impact on patient treatment decisions and outcomes.

We also note that representative microscopic images of the immunohistochemical staining cannot be provided for the present study. The immunohistochemical staining for ER, PR, and Ki67 on the 50 microcores was performed in 2023 at a commercial pathology laboratory operating an automated BenchMark ULTRA platform, under a fee-for-service arrangement common in resource-limited settings where in-house automated IHC is not available. The original stained slides were not archived by the commercial provider beyond their routine retention period, and digital images were not captured at the time of original assessment because the study design relied on direct microscopic scoring by two independent board-certified observers using standardized criteria. The complete quantitative scoring data for each of the 50 microcores, together with the excellent inter-observer agreement (intraclass correlation coefficient = 0.94 for ER, 0.92 for PR, and 0.96 for Ki67), are provided in the S1 Data file. We acknowledge this limitation in the methodological documentation of the present study and recommend that future studies of pre-analytical variation in resource-limited settings incorporate systematic digital image capture as part of the experimental protocol to enable visual reproducibility.

Future research should include multi-center, large-scale multi-specimen studies with diverse tumor types and varying baseline biomarker expression levels to define the relationship between initial biomarker intensity and vulnerability to cold ischemia-induced degradation. Investigation of interventions to mitigate cold ischemia effects (e.g., refrigerated temperatures, specialized transport media) would be valuable. Development of quality control markers indicating adequate pre-analytical handling would be useful. Implementation research is needed to identify effective, sustainable strategies for improving pre-analytical quality in resource-limited settings.

## Conclusion

This exploratory, proof-of-concept controlled experimental study provides preliminary data on pre-analytical handling of breast cancer specimens relevant to resource-limited settings. In a single tumor specimen with strong baseline biomarker expression, non-buffered formaldehyde preserved tissue biomarkers with small but measurable differences relative to neutral buffered formaldehyde for immunohistochemical analysis of ER, PR, and Ki67. While statistically significant differences were observed between fixative types, these did not translate into clinically meaningful classification changes in this strongly-positive specimen, suggesting that cost-effective alternatives to commercial neutral buffered formalin warrant further investigation in multi-specimen studies that include tumors with low or heterogeneous baseline expression.

Cold ischemia time emerged as the critical pre-analytical variable affecting biomarker integrity. Our data strongly support current international guidelines recommending tissue fixation within one hour of specimen collection, as delays beyond two hours resulted in progressive and clinically significant biomarker degradation regardless of fixative type. The preferential loss of progesterone receptor expression with prolonged cold ischemia has particular clinical relevance, as it may lead to hormone receptor status misclassification and inappropriate treatment decisions.

For pathology laboratories in LMICs, these preliminary findings suggest that while some flexibility may exist regarding fixative preparation, rigorous attention to minimizing cold ischemia time remains essential. Implementation of practical workflow improvements—including fixative provision in clinical areas, staff training, efficient specimen transport systems, and cold ischemia time documentation—represents achievable interventions that could improve biomarker assessment quality and reliability. However, these recommendations, while consistent with existing international guidelines, require further validation through multi-center studies with diverse tumor types before being adopted as evidence-based practice changes. Ultimately, optimizing pre-analytical specimen handling is essential for ensuring breast cancer patients in resource-limited settings receive accurate diagnostic information to guide appropriate treatment decisions.

## Supporting information

**S1 Data. Complete dataset including raw biomarker expression data for all fixation duration and cold ischemia time conditions, descriptive statistics, and statistical analysis outputs, including the added Spearman correlations and one-way ANOVA across fixation-duration windows.**
(XLSX)

## Acknowledgments

The authors thank Dr Mwadjie Darolles (Department of Obstetrics and Gynecology, DGOPEH) for providing the mastectomy specimen used in this study. We also thank the surgical and nursing staff of DGOPEH for specimen collection assistance, the technical staff of the Anatomic Pathology department for tissue processing support, and the partner laboratory that provided access to automated immunohistochemistry facilities.

## Author contributions

**Conceptualization:** Clément Parfait Ndengue.

**Data curation:** Gilbert Roger Ateba.

**Formal analysis:** Gilbert Roger Ateba.

**Investigation:** Clément Parfait Ndengue.

**Methodology:** Clément Parfait Ndengue.

**Project administration:** Emile Telesphore MboudoU.

**Resources:** Emile Telesphore MboudoU.

**Software:** Samuel Honoré Mandengue.

**Supervision:** Carole Else Eboumbou Moukoko.

**Validation:** Paul Jean Adrien Atangana.

**Visualization:** Paul Jean Adrien Atangana.

**Writing – original draft:** Clément Parfait Ndengue.

**Writing – review & editing:** Samuel Honoré Mandengue.

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
