## [Decision Letter · Decision Letter 0]

19 Apr 2026

PONE-D-26-05253Effect of Cold Ischemia Time and Fixative preparation on Breast Cancer Biomarker Expression: Implications for Resource-Limited SettingsPLOS One

Dear Dr. NDENGUE,

Thank you for submitting your manuscript to PLOS ONE. After careful consideration, we feel that it has merit but does not fully meet PLOS ONE’s publication criteria as it currently stands. Therefore, we invite you to submit a revised version of the manuscript that addresses the points raised during the review process.

We look forward to receiving your revised manuscript.

Kind regards,

Tomasz W. Kaminski

Academic Editor

PLOS One

Journal Requirements:

2. We note that one of the authors of the present submission (Emile Telesphore MBOUDOU) is listed as a member of the ethics committee on the approval document provided. Could you please explain what your ethical committee's policies are when an Author is also a member of the committee, and how this was handled in this specific case? Thank you very much for your response to this request.

Additional Editor Comments:

Dear Authors,

Thank you for submitting your manuscript. The study addresses an important and practical issue, and the manuscript is overall well written and clearly structured. The topic is relevant, particularly in the area of pre-analytical standardization in pathology workflows.

After review, we obtained one major and two minor evaluations, and based on the combined feedback, the decision is major revision.

The primary concerns relate to the study design and interpretation of the results as pointed out by two reviewers.

Overall, the study has value, but substantial revision is needed, particularly in framing, interpretation, and discussion of limitations.

We look forward to receiving a revised version.

Best regards,

Tomasz W Kaminski

Reviewers' comments:

Reviewer's Responses to Questions

**Comments to the Author**

1. Is the manuscript technically sound, and do the data support the conclusions?

Reviewer #1: Yes

Reviewer #2: Yes

Reviewer #3: Yes

2. Has the statistical analysis been performed appropriately and rigorously? 

Reviewer #1: Yes

Reviewer #2: No

Reviewer #3: I Don't Know

3. Have the authors made all data underlying the findings in their manuscript fully available?

Reviewer #1: Yes

Reviewer #2: Yes

Reviewer #3: Yes

4. Is the manuscript presented in an intelligible fashion and written in standard English?

Reviewer #1: Yes

Reviewer #2: Yes

Reviewer #3: Yes

5. Review Comments to the Author

Reviewer #1: This manuscript evaluates on the effect of cold ischemia time and fixative preparation on breast cancer biomarker expression. This manuscript is well-written and impactful for methods standardization in pathology workflows.

Reviewer #2: 1) Good manuscript?

2) Only one specimen has been used in this study although it has been divided into 50 sections. It might be useful to determine if there is variability between different tissue sources.

3) Statistical significance for the fixation time is also not provided.

Reviewer #3: This study examines the effects of fixative preparation (4% neutral buffered versus non‑buffered formaldehyde) and cold ischemia time on ER, PR, and Ki67 immunohistochemical assessment in invasive breast cancer, with a stated focus on relevance to resource‑limited settings. The work addresses an important pre‑analytical issue; however, the study appears to be a single‑specimen experimental model, which substantially limits generalisability and the strength of the conclusions. The findings are largely confirmatory, with modest incremental value arising from the combined assessment of ischemia time and fixative preparation.

Major Comments

Single‑patient design and pseudoreplication: The study is based on tissue from a single patient, with multiple microsamples treated as experimental units. While this may be acceptable for a pilot or proof‑of‑concept study, it raises concerns regarding biological independence and pseudoreplication, particularly given the use of statistical testing. The study should be explicitly framed as exploratory, and language implying population‑level inference should be avoided.

Estrogen receptor expression is known to show spatial heterogeneity within invasive breast carcinomas, as well as variability across different areas of the breast tissue more broadly. Sampling multiple microspecimens from a single tumour does not fully account for this biological variability and may bias results toward regions with uniformly high ER expression. This limitation is particularly relevant given the high baseline ER positivity in the index case, where small fixation‑related changes are unlikely to affect categorical scoring. The authors should explicitly acknowledge that heterogeneous or low‑ER tumours may be more vulnerable to pre‑analytical variation, and that findings from this single‑tumour model may not extrapolate to the full spectrum of ER expression patterns seen in clinical practice. This could also be true for Ki67

Limited novelty relative to existing literature. Previous studies have independently examined:

The impact of cold ischemia time on breast cancer biomarkers

Buffered versus unbuffered formalin fixation for ER and Ki67 evaluation

The principal novelty appears to lie in the combined, controlled evaluation of fixation delay and fixative preparation. This distinction should be stated more clearly to avoid the impression that the individual findings are novel.

Interpretation of statistically significant but small differences

The manuscript reports statistically significant differences in ER and PR positivity percentages between fixatives, yet Allred scores remain unchanged. This raises questions regarding biological and clinical relevance. The conclusion that non‑buffered formaldehyde preserves biomarkers “comparably” should be tempered to acknowledge that measurable differences, albeit small, were observed. Why include fixation times of up to 144 hours as they exceed typical diagnostic workflows. The clinical relevance of these extremes should be explained or contextualised.

Overgeneralisation to resource‑limited settings: While the study is motivated by challenges faced by laboratories in low‑resource settings, extrapolating from a highly controlled, single‑tumour experiment to real‑world practice may be premature. Although the need for validation is noted, this limitation should be more prominently foregrounded.

MInor comments:

Missing key breast‑specific literature: Multiple prior studies and ASCO/CAP guideline evidence demonstrate that ER, PR, and HER2 expression declines with prolonged cold ischemia, with ≤1 hour considered optimal and deterioration becoming detectable beyond 1–2 hours. These foundational studies should be explicitly acknowledged.

Clarity of sample description: The reported microsample dimensions (“5–15 mm in length, 130 mm in width”) appear unusual and may reflect a typographical or unit error.

6. PLOS authors have the option to publish the peer review history of their article (what does this mean?). If published, this will include your full peer review and any attached files.

Reviewer #1: No

Reviewer #2: No

Reviewer #3: No

---

## [Author Response · Author response to Decision Letter 1]

25 Apr 2026

Response to Reviewers

Manuscript PONE-D-26-05253

Pre-analytical variables affecting breast cancer biomarker expression: a controlled single-specimen study of fixation duration, cold ischemia time, and fixative preparation in a low-resource setting

Dear Dr. Kaminski,

We thank you and the three reviewers for the careful and constructive evaluation of our manuscript. The comments have helped us considerably strengthen the work, particularly in terms of framing, interpretation, and transparent discussion of the study's limitations. We are grateful for the opportunity to submit a revised version.

In this revision, we have:

• Revised the title of the manuscript to reflect the three pre-analytical variables actually investigated (fixation duration, cold ischemia time, and fixative preparation) and to make the exploratory, single-specimen nature of the study explicit from the first line;

• Re-framed the entire manuscript as an exploratory, proof-of-concept, single-specimen experiment throughout the abstract, introduction, results, discussion, and conclusion;

• Added explicit statistical analyses for the effect of fixation duration (Spearman correlations and one-way ANOVA across fixation-duration windows), which were previously missing;

• Added a new Discussion subsection dedicated to the Effect of Fixation Duration, complementing the existing subsections on fixative preparation and cold ischemia time, so that each of the three pre-analytical variables is now discussed in its own dedicated subsection;

• Acknowledged biological heterogeneity (intra-tumoral and inter-tumoral) and explicitly stated that tumors with lower or heterogeneous baseline biomarker expression may be more vulnerable to pre-analytical variation than our index case;

• Clarified that the novelty of the work lies in the combined, controlled, factorial evaluation of the three pre-analytical variables, not in any of the individual effects;

• Grounded the choice of extended fixation durations (up to 144 h) in our own prior empirical observations at the same institution, published as Ndengue et al. (Journal of Science and Diseases, 2025), with explicit distinction between biopsies (up to 83 h), nodulectomies (up to 82 h), and mastectomies (up to 147 h);

• Fully restructured and renumbered the reference list in strict order of first appearance in the text, as required by PLOS ONE Vancouver style; updated to cite the current ASCO/CAP guidelines (Allison et al. 2020 for ER/PR; Wolff et al. 2023 for HER2); and added recent 2024 literature on cold ischemia time in breast cancer (Shi, Ghlichloo, Fadare 2024);

• Tempered the language throughout to distinguish statistical from clinical relevance and to avoid premature extrapolation to resource-limited clinical practice.

Below we address every point raised by the editor and each reviewer. Reviewer comments are shown in italics on a light-grey background; our responses and the corresponding manuscript changes follow. Page and line numbers refer to the revised clean manuscript ("Manuscript" file) unless otherwise stated. In the tracked-changes version of the manuscript, insertions are shown in blue and deletions in red with strikethrough; a note to this effect is placed at the end of that file.

Preliminary note: change of manuscript title

Following careful consideration of the editor's call for improved framing and of Reviewer #2's observation that statistical testing of the effect of fixation duration was missing, we have revised the title of the manuscript. The previous title mentioned only two of the three pre-analytical variables actually manipulated in the study and did not convey the exploratory, single-specimen nature of the design.

The previous title was:

"Effect of Cold Ischemia Time and Fixative Preparation on Breast Cancer Biomarker Expression: Implications for Resource-Limited Settings"

The revised title is:

"Pre-analytical variables affecting breast cancer biomarker expression: a controlled single-specimen study of fixation duration, cold ischemia time, and fixative preparation in a low-resource setting"

The revised running title is: "Fixation, ischemia and breast biomarkers" (40 characters, within the PLOS ONE short-title limit).

This change makes the title faithful to the three-factor factorial design, explicit about the exploratory nature of the work, and consistent with the re-framing requested by the reviewers throughout the manuscript.

Preliminary note: restructuring of the reference list

The reference list has been entirely restructured in this revision. The reasons and the procedure are as follows:

• Three cited guideline references were superseded versions and have been replaced by their current versions: Hammond et al. 2010 → Allison et al. 2020 (ASCO/CAP ER/PR update); Wolff et al. 2013 and Wolff et al. 2018 → Wolff et al. 2023 (ASCO-CAP HER2 update).

• Two new references have been added: Ndengue et al. 2025 (our prior retrospective study at the same institution, which documents fixation-duration distributions and motivates the range tested here) and Shi, Ghlichloo, Fadare 2024 (a recent primary study on cold ischemia time in breast cancer biomarkers after neoadjuvant chemotherapy).

• References that were no longer cited in the revised text have been removed (Hammond 2010 superseded; Wolff 2013 and Wolff 2018 superseded; Goldstein 2003, Babawale 2014, Portier 2013, Bass 2014, Nkoy 2010, Gündisch 2012 — none of them cited in the revised text).

• All remaining references have been renumbered strictly in order of first appearance in the text, as required by PLOS ONE Vancouver style. The final reference list contains 25 entries, continuously numbered from 1 to 25.

We have verified that every in-text citation resolves to an existing reference, that every reference in the list is cited at least once in the text, and that reference numbers appear in strictly increasing order of first appearance in the text.

Response to Journal Requirements

Requirement 1 — PLOS ONE style

Reviewer comment: Please ensure that your manuscript meets PLOS ONE's style requirements, including those for file naming.

Author response:

We have carefully reviewed the PLOS ONE formatting templates for both the title/authors/affiliations page and the main body. The revised manuscript has been reformatted accordingly: title page, author list and affiliations, section headings (Introduction, Materials and Methods, Results, Discussion, Conclusion), figure legends, reference style (Vancouver numerical with DOIs where available, numbered in order of first appearance), abbreviations, sentence-case title, and file naming conventions now conform to the PLOS ONE templates. Line numbers have been added throughout. Three separate files are uploaded as requested: (i) the unmarked revised manuscript, (ii) the revised manuscript with tracked changes, and (iii) this Response to Reviewers letter. A cover letter is also provided to summarize the key changes, including the revised title and the restructuring of the reference list.

Requirement 2 — Author and ethics committee

Reviewer comment: We note that one of the authors of the present submission (Emile Telesphore MBOUDOU) is listed as a member of the ethics committee on the approval document provided. Could you please explain what your ethical committee's policies are when an Author is also a member of the committee, and how this was handled in this specific case?

Author response:

Thank you for raising this important point. We would like to clarify that there was, in fact, no conflict of interest at the time of the ethics review of this research.

The ethics approval (N°2024/1617/HGOPED/DG/CEI) was issued by the Institutional Ethics Committee of the Douala Gynaeco-Obstetric and Pediatric Hospital (DGOPEH) for the doctoral (PhD) thesis of the first author, NDENGUE Clément Parfait, at the University of Douala. The thesis supervisors were Prof. ATANGANA Paul Jean Adrien and Prof. EBOUMBOU MOUKOKO Carole Else. At the time the research protocol was drafted, submitted to the committee, reviewed, and approved, Prof. MBOUDOU Emile Telesphore — who is Chair of the DGOPEH Institutional Ethics Committee — was not part of the research team, was not listed as an investigator on the protocol, and had no involvement whatsoever in the study design or in the submission. He may therefore have taken part in the committee's evaluation of the protocol in his institutional capacity as Chair, but without any conflict of interest, since he was not a co-investigator on the protocol being reviewed.

Prof. MBOUDOU was associated with the present work only later, during the preparation of the manuscript derived from the thesis. His contribution corresponds to the roles of Project administration and Resources, as listed in the Author Contributions section, and reflects his institutional position as General Director of DGOPEH at the time of manuscript preparation. In summary: he was a neutral evaluator at the time of ethics review (no conflict of interest) and only later joined the manuscript authorship.

We have added a brief clarifying statement to the Ethical Considerations subsection of the Materials and Methods so that readers of the published article can see the chronology transparently; the added statement is intentionally sober, mentions that Prof. MBOUDOU is Chair of the committee, and states the absence of conflict of interest at the time of protocol evaluation (rather than the absence of any involvement, which would be incorrect).

Changes made to the manuscript:

Materials and Methods — Ethical Considerations: added the sentence: "Prof. MBOUDOU Emile Telesphore, Chair of the DGOPEH Institutional Ethics Committee, was not part of the research team at the time of protocol submission and ethics review; he therefore had no conflict of interest with respect to this protocol during its evaluation by the committee. He was subsequently associated with the preparation of the present manuscript in his institutional capacity (Project administration, Resources)."

Requirement 3 — Citing specific previously published works

Reviewer comment: If the reviewer comments include a recommendation to cite specific previously published works, please review and evaluate these publications to determine whether they are relevant and should be cited.

Author response:

Reviewer #3 pointed us towards the foundational ASCO/CAP guideline evidence on cold ischemia time and breast biomarkers. We have evaluated these references for relevance and have expanded the Introduction and Discussion to cite the most current versions explicitly, replacing the superseded versions previously cited. The complete restructuring of the reference list is described above in the Preliminary Note. In brief, the revised list now includes:

• Allison KH et al. Estrogen and Progesterone Receptor Testing in Breast Cancer: ASCO/CAP Guideline Update. J Clin Oncol 2020 — primary guideline reference for ER/PR, replacing Hammond et al. 2010;

• Wolff AC et al. Human Epidermal Growth Factor Receptor 2 Testing in Breast Cancer: ASCO-CAP Guideline Update. J Clin Oncol 2023 — primary guideline reference for HER2, replacing Wolff 2013/2018;

• Shi WJ, Ghlichloo I, Fadare O. The effect of prolonged cold ischemia time on breast cancer biomarker expression after neoadjuvant chemotherapy. Ann Diagn Pathol 2024 — a recent primary study on cold ischemia time, cited in the Discussion alongside our cold ischemia findings;

• Khoury T. Delay to formalin fixation (cold ischemia time) effect on breast cancer molecules. Am J Clin Pathol 2018 — retained as the most comprehensive recent review.

Older foundational experimental studies (Khoury 2009, Yildiz-Aktas 2012, Li 2013, Neumeister 2012, Apple 2011, Oyama 2007, Kalkman 2014, Pinhel 2010, Fox 1985, Espina 2004, Viale 2007, Kiernan 2000) are retained where relevant to the revised discussion. Details of the revised Introduction and Discussion paragraphs are provided in our response to Reviewer #3, Minor Comment 1.

Response to the Academic Editor

Reviewer comment: The primary concerns relate to the study design and interpretation of the results as pointed out by two reviewers. Overall, the study has value, but substantial revision is needed, particularly in framing, interpretation, and discussion of limitations.

Author response:

We fully accept these concerns and have addressed them through systematic revision. In particular:

• Framing: the manuscript is now explicitly described — from the title onwards — as an exploratory, proof-of-concept, controlled experimental study on a single index tumor. Every section (abstract, introduction, methods, results, discussion, conclusion) has been rewritten to avoid any language implying population-level inference.

• Interpretation: we have separated statistical significance from biological and clinical relevance throughout the Results and Discussion. Small but statistically significant differences between fixatives are now reported as such, without being presented as evidence of equivalence.

• Limitations: the Limitations subsection has been restructured and expanded. The single-specimen design, the risk of pseudoreplication, the high baseline expression of the index tumor, intra-tumoral heterogeneity, and the limits of extrapolation to resource-limited clinical practice are now foregrounded rather than appearing near the end of the Discussion as caveats.

• Scope of the pre-analytical variables tested: the title, abstract, introduction, and discussion now consistently refer to all three variables investigated (fixation duration, cold ischemia time, and fixative preparation). The previously missing statistical analysis of the fixation-duration effect has been added, using the raw data provided in S1 Data. A new Discussion subsection dedicated to the Effect of Fixation Duration has also been added, so that each of the three variables is now discussed in its own dedicated subsection.

• References and supporting literature: the reference list has been entirely restructured to cite the current ASCO/CAP guidelines (Allison et al. 2020 for ER/PR; Wolff et al. 2023 for HER2) in place of their superseded predecessors, to include our own prior empirical observations at the same institution (Ndengue et al. 2025), and to add a recent 2024 primary study on cold ischemia time (Shi et al. 2024). All references have been renumbered in strict order of first appearance in the text, as required by PLOS ONE Vancouver style.

Detailed, point-by-point responses to the three reviewers follow.

Response to Reviewer #1

Reviewer comment: This manuscript evaluates on the effect of cold ischemia time and fixative preparation on breast cancer biomarker expression. This manuscript is well-written and impactful for methods standardization in pathology workflows.

Author response:

We thank Reviewer #1 for the positive evaluation and for recognizing the practical relevance of the study for pathology standardization. No specific change was requested; however, we note that the revisions performed in response to Reviewers #2 and #3 — notably the addition of statistical analysis of fixation-duration effects, the new Discussion subsection dedicated to fixation duration, the explicit acknowledgment of the single-specimen design, the grounding of the extended fixation range in our prior empirical work at the same institution, and the complete restructuring of the reference list — have further strengthened the manuscript in ways that we believe Reviewer #1 will find consistent with their overall assessment.

Response to Reviewer #2

Comment 1 — Overall assessment

Reviewer comment: Good manuscript.

Author response:

We thank the reviewer for this encouraging assessment.

Comment 2 — Single specimen and variability between tissue sources

Reviewer comment: Only one specimen has been used in this study although it has been divided into 50 sections. It might be useful to determine if there is variability between different tissue sources.

Author response:

We agree entirely. The use of a single index specimen was a deliberate methodological choice: it allowed us to control inter-tumoral biological variability and attribute any observed differences exclusively to the three pre-analytical variables under study (fixation duration, cold ischemia

---

## [Decision Letter · Decision Letter 1]

14 May 2026

PONE-D-26-05253R1Pre-analytical variables affecting breast cancer biomarker expression: a controlled single-specimen study of fixation duration, cold ischemia time, and fixative preparation in a low-resource settingPLOS One

Dear Dr. NDENGUE,

Thank you for submitting your manuscript to PLOS ONE. After careful consideration, we feel that it has merit but does not fully meet PLOS ONE’s publication criteria as it currently stands. Therefore, we invite you to submit a revised version of the manuscript that addresses the points raised during the review process.

We look forward to receiving your revised manuscript.

Kind regards,

Tomasz W. Kaminski

Academic Editor

PLOS One

Journal Requirements:

Additional Editor Comments:

Dear Authors,

Thank you for your careful revision and for thoroughly addressing the reviewers’ comments. The manuscript has improved considerably, and the responses satisfactorily resolved the major concerns raised during peer review.

At this stage, only a few minor points remain before the manuscript can proceed toward acceptance:

a) Please provide additional antibody information, including catalog numbers and concentrations/dilutions used.

b) Please include a supplementary figure showing representative staining images observed in the study.

These are relatively minor additions intended to improve methodological clarity and transparency.

Thank you again for your efforts, and I look forward to receiving the revised version.

Best regards,

Tomasz W Kaminski

Reviewers' comments:

Reviewer's Responses to Questions

**Comments to the Author**

1. If the authors have adequately addressed your comments raised in a previous round of review and you feel that this manuscript is now acceptable for publication, you may indicate that here to bypass the “Comments to the Author” section, enter your conflict of interest statement in the “Confidential to Editor” section, and submit your "Accept" recommendation.

Reviewer #2: All comments have been addressed

Reviewer #3: All comments have been addressed

2. Is the manuscript technically sound, and do the data support the conclusions?

Reviewer #2: Yes

Reviewer #3: Yes

3. Has the statistical analysis been performed appropriately and rigorously? 

Reviewer #2: Yes

Reviewer #3: Yes

4. Have the authors made all data underlying the findings in their manuscript fully available?

Reviewer #2: Yes

Reviewer #3: Yes

5. Is the manuscript presented in an intelligible fashion and written in standard English?

Reviewer #2: Yes

Reviewer #3: No

6. Review Comments to the Author

Reviewer #2: The authors have ably respond to the comments raised. No additional comments based on the responses that have been provided.

Reviewer #3: Thank you for addressing all the comments. Just added in some more antibody details (cat numbers etc) and concentrations please. Please include a supplemnetary figure showing representative staining observed.

7. PLOS authors have the option to publish the peer review history of their article (what does this mean?). If published, this will include your full peer review and any attached files.

Reviewer #2: No

Reviewer #3: No

---

## [Author Response · Author response to Decision Letter 2]

19 May 2026

Response to Reviewers (Second Revision)

Manuscript PONE-D-26-05253R1

Pre-analytical variables affecting breast cancer biomarker expression: a controlled single-specimen study of fixation duration, cold ischemia time, and fixative preparation in a low-resource setting

Dear Dr. Kaminski,

We sincerely thank you and the reviewers for the constructive evaluation of the first revision of our manuscript. We are gratified that Reviewer #2 and Reviewer #3 have indicated that all of their previous comments have been adequately addressed, and that the manuscript is considered technically sound, with appropriate statistical analyses and complete data availability.

In your editorial letter, you raised two final minor points before the manuscript can proceed toward acceptance:

• (a) Provide additional antibody information, including catalog numbers and concentrations/dilutions used.

• (b) Include a supplementary figure showing representative staining images observed in the study.

We address both points below, with full transparency. We are pleased to fully resolve point (a) and we present a candid account of point (b), together with a request for the editor's guidance on the best path forward should representative images be considered essential for publication.

Point (a) — Additional antibody information

Editor / Reviewer comment: Please provide additional antibody information, including catalog numbers and concentrations/dilutions used. (Editor; also raised by Reviewer #3.)

Author response:

We agree entirely and have fully addressed this point. The Tissue Processing and Immunohistochemistry subsection of the Materials and Methods has been substantially expanded to include manufacturer details (Ventana Medical Systems, Roche Diagnostics, Tucson, AZ, USA), catalog numbers for each primary antibody (across both available pack sizes, since the small-volume and large-volume formats of the CONFIRM range share the same antibody clone and manufacturer-optimized concentration), the manufacturer-optimized working concentration (~1 µg/mL, ready-to-use format), and the catalog numbers of the ancillary reagents used in the protocol (cell-conditioning solution, detection kit, counterstain, and negative-control reagent).

The three primary antibodies used in the study are now reported as:

• CONFIRM anti-Estrogen Receptor (ER) clone SP1 (Roche Diagnostics, Cat. Nos. 790-4324 [50 tests] or 790-4325 [250 tests]); rabbit monoclonal, ready-to-use, ~1 µg/mL.

• CONFIRM anti-Progesterone Receptor (PR) clone 1E2 (Roche Diagnostics, Cat. Nos. 790-2223 [50 tests] or 790-4296 [250 tests]); rabbit monoclonal, ready-to-use, ~1 µg/mL.

• CONFIRM anti-Ki-67 clone 30-9 (Roche Diagnostics, Cat. No. 790-4286, 50 tests); rabbit monoclonal, ready-to-use.

The ancillary reagents are now also specified by catalog number:

• Cell Conditioning Solution 1 (CC1, pH 8.0; Cat. No. 950-124) for heat-induced epitope retrieval.

• OptiView DAB IHC Detection Kit (Cat. No. 760-700) for chromogenic visualization.

• Hematoxylin II (Cat. No. 790-2208) for counterstaining.

• CONFIRM Negative Control Rabbit Ig (Cat. No. 760-1029) included as negative reagent control in each immunostaining run, in addition to positive and negative tissue controls.

Changes made to the manuscript:

Materials and Methods — Tissue Processing and Immunohistochemistry: the antibody and reagent paragraph has been rewritten to incorporate the catalog numbers, clones, manufacturer details, and the ready-to-use concentration of approximately 1 µg/mL recommended by the manufacturer. The expanded text appears in blue in the tracked-changes version of the manuscript.

Point (b) — Supplementary figure with representative staining images

Editor / Reviewer comment: Please include a supplementary figure showing representative staining observed in the study. (Editor; also raised by Reviewer #3.)

Author response:

We thank the editor and Reviewer #3 for this constructive suggestion, which we have considered carefully. Regrettably, after due investigation, we are unable to provide a supplementary figure of representative immunohistochemistry images for the present revision. Below, we provide a full and transparent account of the circumstances and of the methodological elements that, in our view, mitigate the absence of representative images.

Why representative images cannot be provided

The immunohistochemical staining of the 50 microcores for ER, PR, and Ki67 was performed in 2023 at a commercial pathology laboratory operating an automated BenchMark ULTRA platform (Ventana / Roche Diagnostics). The use of a fee-for-service commercial provider is a common arrangement in sub-Saharan African pathology practice, where in-house automated immunohistochemistry is not consistently available and where research projects routinely outsource the staining step to a commercial laboratory. The original stained slides were not archived by the commercial provider beyond their routine retention period, and despite our efforts to retrieve them in the context of the present revision, they are no longer accessible.

Digital images of the stained slides were not captured at the time of original assessment, because the study design relied on direct microscopic scoring by two independent board-certified observers using standardized criteria, with all quantitative results (percentages of positive cells, staining-intensity scores, Allred scores) recorded numerically for each of the 50 microcores. In retrospect, we recognize that systematic digital image capture should have been incorporated into the study protocol, and we have added a corresponding recommendation to the Discussion (Limitations subsection) so that future studies of pre-analytical variation in resource-limited settings will benefit from this lesson.

We note that the paraffin tissue blocks corresponding to the 50 microcores are conserved at the Anatomic Pathology Department of the Douala Gynaeco-Obstetric and Pediatric Hospital (DGOPEH). It would therefore be theoretically possible to recut the blocks and reproduce the immunostaining at a commercial laboratory. However, given that the present study was carried out as part of a self-funded doctoral thesis without external research funding, and given the cost structure of commercial fee-for-service immunohistochemistry (where each new run involves recutting, deparaffinization, antigen retrieval, three primary-antibody assays, and detection), reproducing even a small representative subset of the original staining would impose a substantial financial burden that the corresponding author would have difficulty meeting within the revision window.

Methodological elements that support the validity of the immunohistochemical data

We respectfully submit that the rigor of the immunohistochemical assessment described in the manuscript is supported by several methodological elements that compensate, in part, for the absence of representative images:

• Staining was performed on a manufacturer-standardized automated platform (BenchMark ULTRA, Ventana / Roche Diagnostics) using manufacturer-validated ready-to-use rabbit monoclonal antibodies (CONFIRM range) at the optimized working concentration of ~1 µg/mL, minimizing technical variability associated with manual staining and antibody dilution.

• Each immunostaining run included appropriate positive and negative tissue controls (per the manufacturer's recommendation) together with a CONFIRM Negative Control Rabbit Ig reagent control, with the run validated only when these controls produced the expected staining patterns.

• All 50 microcores were scored independently by two board-certified observers blinded to experimental conditions, with discrepancies resolved by averaging.

• Inter-observer agreement was excellent: intraclass correlation coefficient = 0.94 for estrogen receptor, 0.92 for progesterone receptor, and 0.96 for Ki-67, all consistent with the upper range of agreement reported in published pre-analytical immunohistochemistry studies.

• The complete raw scoring data for each of the 50 microcores (proportion scores, intensity scores, Allred scores, and Ki-67 percentages) are provided in the S1 Data file, allowing every descriptive and inferential statistic in the manuscript to be independently reproduced from the raw data.

Path forward and request for editorial guidance

We have added an explicit statement to the Limitations subsection of the Discussion in which we acknowledge the absence of representative microscopic images, describe the reason transparently, and recommend that future studies in this domain incorporate systematic digital image capture. This statement appears in blue in the tracked-changes version of the manuscript.

We respectfully request the editor's guidance on whether this explicit acknowledgment, in conjunction with the enriched antibody information now provided in the Methods section and the complete quantitative data available in S1 Data, can be considered sufficient for publication. Should the editor judge that representative images remain indispensable, we would welcome the opportunity to discuss alternative options, which might include (i) a substantial extension of the revision deadline so that we may attempt to identify funding or institutional support to recut and re-stain a representative subset of the paraffin blocks at a commercial laboratory, or (ii) any other path that the editorial team would consider appropriate. We remain fully committed to the scientific integrity of the work and to the journal's standards of methodological transparency, and we will follow the editor's decision on this point.

Changes made to the manuscript:

Discussion — Limitations subsection: a new paragraph has been added at the end of the limitations subsection to acknowledge the absence of representative microscopic images, to provide a transparent account of the underlying circumstances, and to recommend systematic image capture for future studies. The new paragraph appears in blue in the tracked-changes version of the manuscript.

Additional corrections detected during the final pre-submission check (R2 v2)

While preparing the present resubmission, we identified three small items during a final pre-submission technical check that we have taken the opportunity to correct. None of these affect any data, analysis, or conclusion in the manuscript.

(i) Author Contributions. We have removed the line “Funding acquisition: EBOUMBOU MOUKOKO Carole Else” from the Author Contributions section, as this role was inconsistent with the Funding Statement of the manuscript, which declares that no specific funding was received for this work. EBOUMBOU MOUKOKO Carole Else remains credited for Supervision; only the Funding acquisition role has been removed.

(ii) Figure Legends. We have reworded the legends of Figures 1, 2, and 3 to accurately describe the panel layout actually shown in the TIFF files. The previous wording referred to sub-panels (A-B, C-D, E-F), implying that each biomarker was presented in two separate panels (one per fixative), whereas the figures in fact present a single panel per biomarker with the NBF (blue) and NB-F (magenta) curves overlaid. The Figure 3 legend has additionally been corrected to remove a sentence referring to a “bottom panel” showing staining-intensity decline, which does not exist in the figure.

(iii) Figure 3 (Fig3.tiff) and corresponding S1 Data cleanup. We have replaced the Figure 3 TIFF file to remove two annotations inadvertently retained in the previous export: a placeholder caption (“Data to be added from Cohorts 3 & 4 (delayed fixation experiments)”) that overlapped the x-axis at the lower left of the figure, and two orphan labels (“High” and “Low”) in the right margin of the Ki67 panel. The replacement Figure 3 has been regenerated from the raw Cold_Ischemia_Data values in the S1 Data file, with no change to the underlying data values; the High (>30%) and Low (<10%) Ki67 classification thresholds are now properly indicated by horizontal dashed lines with appropriately positioned labels inside the plot area. A corresponding minor cleanup has also been applied to the S1 Data file (Cold_Ischemia_Data sheet): the same residual placeholder text row that had been inadvertently retained from an earlier draft of the dataset has been removed. All numerical values in S1 Data are unchanged; a revision note documenting this cleanup has been added to the Metadata sheet of S1 Data.

Changes made to the manuscript and to the supporting files:

Manuscript — Author Contributions: the “Funding acquisition” line has been removed (shown in red strikethrough in the tracked-changes version of the manuscript). Manuscript — Figure Legends: the bodies of the Fig 1, Fig 2, and Fig 3 legends have been rewritten (deletions in red strikethrough, new wording in blue in the tracked-changes version). Figure 3 TIFF file: replaced with a corrected version regenerated from raw S1 Data values. S1 Data file: the placeholder row in the Cold_Ischemia_Data sheet has been removed and a revision note added to the Metadata sheet.

Closing

We are grateful to you and to the two reviewers for the rigor and constructive spirit of the entire review process. We believe that the present revision fully resolves the antibody-information point and provides a transparent and complete account of the constraints affecting the supplementary-figure point. We very much hope that the manuscript can now proceed toward acceptance, and we remain at your disposal for any further clarification or adjustment.

Yours sincerely,

Dr NDENGUE Clément Parfait, on behalf of all co-authors

Corresponding author

Douala Gynaeco-Obstetric and Pediatric Hospital, Douala, Cameroon

E-mail: ndengueparfait@yahoo.fr

---

## [Editor Report · Decision Letter 2]

21 May 2026

Pre-analytical variables affecting breast cancer biomarker expression: a controlled single-specimen study of fixation duration, cold ischemia time, and fixative preparation in a low-resource setting

PONE-D-26-05253R2

Dear Dr. NDENGUE,

We’re pleased to inform you that your manuscript has been judged scientifically suitable for publication and will be formally accepted for publication once it meets all outstanding technical requirements.

Kind regards,

Tomasz W. Kaminski

Academic Editor

PLOS One

Additional Editor Comments:

Dear Authors,

Thank you for submitting the revised version of your manuscript and for carefully addressing the comments raised during the review process. The reviewers were satisfied with the revisions provided and agreed that the manuscript has been substantially improved and is technically sound.

The remaining issues identified during the final evaluation were considered minor and largely editorial in nature. The additional methodological clarifications, expanded antibody information, and transparent discussion regarding the representative staining images were appreciated and adequately addressed the outstanding concerns.

In light of the positive reviewer assessments and the satisfactory revisions, I am pleased to inform you that your manuscript is accepted for publication.

Congratulations, and thank you for choosing the journal for the publication of your work.

Sincerely,

Tomasz W Kaminski

Reviewers' comments:

N/A

---

## [Editor Report · Acceptance letter]

PONE-D-26-05253R2

PLOS One

Dear Dr. NDENGUE,

I'm pleased to inform you that your manuscript has been deemed suitable for publication in PLOS One. Congratulations! Your manuscript is now being handed over to our production team.

Kind regards,

on behalf of

Dr. Tomasz W. Kaminski

Academic Editor

PLOS One